# Gut Microbiota and Dietary Factors as Modulators of the Mucus Layer in Inflammatory Bowel Disease

**DOI:** 10.3390/ijms221910224

**Published:** 2021-09-23

**Authors:** Samuel Fernández-Tomé, Lorena Ortega Moreno, María Chaparro, Javier P. Gisbert

**Affiliations:** 1Gastroenterology Unit, Hospital Universitario de La Princesa, Instituto de Investigación Sanitaria Princesa (IIS-IP), Universidad Autónoma de Madrid (UAM), 28006 Madrid, Spain; lorena.ortega8317@gmail.com (L.O.M.); mariachs2005@gmail.com (M.C.); javier.p.gisbert@gmail.com (J.P.G.); 2Centro de Investigación Biomédica en Red de Enfermedades Hepáticas y Digestivas (CIBEREHD), 28006 Madrid, Spain

**Keywords:** dietary compounds, gastrointestinal barrier, gut microbiota, inflammatory bowel disease, mucus layer

## Abstract

The gastrointestinal tract is optimized to efficiently absorb nutrients and provide a competent barrier against a variety of lumen environmental compounds. Different regulatory mechanisms jointly collaborate to maintain intestinal homeostasis, but alterations in these mechanisms lead to a dysfunctional gastrointestinal barrier and are associated to several inflammatory conditions usually found in chronic pathologies such as inflammatory bowel disease (IBD). The gastrointestinal mucus, mostly composed of mucin glycoproteins, covers the epithelium and plays an essential role in digestive and barrier functions. However, its regulation is very dynamic and is still poorly understood. This review presents some aspects concerning the role of mucus in gut health and its alterations in IBD. In addition, the impact of gut microbiota and dietary compounds as environmental factors modulating the mucus layer is addressed. To date, studies have evidenced the impact of the three-way interplay between the microbiome, diet and the mucus layer on the gut barrier, host immune system and IBD. This review emphasizes the need to address current limitations on this topic, especially regarding the design of robust human trials and highlights the potential interest of improving our understanding of the regulation of the intestinal mucus barrier in IBD.

## 1. Introduction to Inflammatory Bowel Disease

Inflammatory bowel disease (IBD) is a global disease associated to Western and recently westernized countries [1]. The emergence of this disease was parallel to the industrial revolution in the 1800s [2]. Being a chronic disease diagnosed early in life, the prevalence of this pathology is high and is increasing over time. Prevalence of IBD was 84 per 100,000 population in 2017 [3] and it has been estimated that it will continue increasing in the next generation, affecting tens of millions of people all over the world [4]. Therefore, the cost of this disease for health care systems is considerable and will increase steadily in the future [4,5].

The origin and causes of IBD remain unknown. It is an immune-mediated inflammatory disease and its major causative factors could be genetic, immune and environmental such as the gut microbiome and diet. Genome wide-association studies identified approximately 200 gene loci in IBD, of which more than 50% are also associated with other inflammatory and autoimmune diseases [6]. The exposure to environmental conditions influence the microbiome composition and the consequent dysbiosis (changes in the healthy microbiota) in the gastrointestinal tract can trigger inflammatory responses [7,8].

IBD is a general term encompassing ulcerative colitis (UC) and Crohn’s disease (CD). UC is limited to the colon and presents superficial mucosal inflammation that can lead to ulcerations and bleeding. CD can affect any part of the digestive tract and presents transmural inflammation and complications such as fistulas or abscesses [9]; furthermore, IBD is associated to other extra-intestinal pathologies such as arthritis and skin diseases that aggravate the quality of life of these patients. Both IBD subtypes present periods of inflammation and quiescence [10]. Regarding IBD therapeutic approaches, several drugs have been developed over the last years, including biologics that target different molecules involved in IBD pathogenesis [11,12]. However, response to treatment is highly variable [13,14] and, since there is no cure for this disease, the therapeutic goal is to maintain patients’ remission. Accordingly, a deeper understanding of the disease is needed to improve treatment of these patients.

In this review, we will focus on the gastrointestinal barrier in IBD with a particular emphasis on the role of the mucus layer in gut health and its alterations in this disease. In addition, the impact of the gut microbiota and dietary compounds as mucus modulatory factors and their complex interaction with the mucosal barrier in IBD is summarized. Data were obtained from articles published in English in journals indexed in PubMed and Web of Science from inception to August 2021 and retrieved using search terms related to (i) gastrointestinal barrier and gut homeostasis; (ii) mucus layer, mucins and IBD; (iii) modulation of immune system and mucosal inflammation; (iv) gut microbiota, probiotics and IBD; (v) dietary compounds, food bioactives and IBD.

## 2. Gastrointestinal Barrier

The intestinal mucosal barrier provides adequate containment of microorganisms and molecules, preserving the capacity to absorb nutrients [15]. Intestinal mucosa is covered with a monolayer of intestinal epithelial cells (IECs) that separate the external environment and sub-epithelium [16]. Alterations in this mucosal barrier may result in IBD, stressing its essential role to maintain a healthy gut environment [17]. A key regulator balancing this relationship is the gastrointestinal mucus layer, composed of a secreted mucus gel, which cover the surface of epithelium and the underlying mucosal immune system. Hence, the gut mucosa is protected by two barrier types: chemical and physical. Chemical barriers participate in the segregation of IECs and gut microbiota [18]. IECs are derived from stem cells within intestinal crypts that replicate and migrate towards villi to replenish the active turnover of epithelium [15]. Functionally, secretory IECs, as goblet and Paneth cells, are specialized in maintaining the epithelial barrier function [19]. Paneth cells are involved in the production of chemical barriers such as antimicrobial peptides in the small intestine [20], while goblet cells secrete mucins. Mucins and antimicrobial peptides are important for both physical and biochemical barriers. The different functions of IECs lead to a dynamic barrier, which protects the host from infection and inflammatory stimuli [19]. IECs act as sensors for microbial elements and can integrate signals from commensal bacteria into antimicrobial and immune regulatory responses [21]. These functions are enabled by the expression of pattern-recognition receptors that act as sensors of the microbial environment and are key regulatory elements in mucosal immune responses [19].

Mucosal homeostasis is a vital feature of the gut immune system [22]. One of the critical factors for developing IBD is the failure to maintain an adequate balance between response to pathogens and tolerance to commensal microorganisms and luminal beneficial antigens [23,24]. Under the conditions of gut barrier dysfunction, as it occurs in IBD, the homeostatic equilibrium is lost [25,26]. IBD is related with increased permeability in the gut and the associated disbalance in the immune response that leads to increased recruitment of circulating cells and secretion of pro-inflammatory mediators [15,27]. Therefore, factors as immune system, genetics and environmental ones influence the gastrointestinal barrier function and are, thus, involved in the “IBD integrome” [28].

### 2.1. Mucus Layer

The small intestine has a single mucus layer that facilitates the pass of nutrients, while the colon is covered by a thicker barrier. However, in the colon, the mucus layer acts as a physical barrier maintaining bacteria in symbiosis with the host and preventing bacterial infiltration into the epithelium [16,18]. The large intestine epithelium is, thus, covered by two mucus layers: an outer loose layer and an inner firm mucus attached to the epithelia [29,30]. The principal components of the gastrointestinal mucus barrier are O-linked glycoproteins called mucins. They present densely packed oligosaccharides that bind to their terminal region sialic acid and sulfate residues protecting mucins from proteases and glycosidases [31]. Mucins are produced by goblet cells present within the intestinal epithelium [32]. Mucus exocytosis from goblet cells depends on several cellular processes that modulate mucin secretion, including endocytosis and autophagy [32].

There are 18 mucin members in humans classified in two types: transmembrane and secreted mucins. Mucin central domains are composed of proline, threonine and serine (PTS) residues working as attachment sites for O-linked glycans through covalent binding of N-acetylgalactosamine to serine or threonine residues [16]. The secreted mucin MUC2 is the main glycoprotein in the intestinal mucus. MUC2 has an N-terminal domain, two PTS domains and a C-terminal domain. MUC2 N-terminal domain comprises 3 complete von Willebrand factor domains (D1-3) and the C-terminal region of D4 domain. Cysteine residues in N- and C- terminal domains facilitate inter- and intramolecular disulfide bond formation responsible for mucin polymerization [33].

MUC2 polypeptide is synthetized and dimerized in the endoplasmic reticulum of intestinal cells. Then, threonine and serine residues are glycosylated in the cis-Golgi and the trimer formation takes place in the trans-Golgi before MUC2 is packaged into secretory granules. MUC2 is composed of heterogeneous glycan chains [16], which allow MUC2 trimers to form polymers creating mucus networks in the cell surface [31,34]. MUC2 polymers undergo rapid expansion on the intestinal epithelial surface to maintain the mucus barrier during homeostasis; this expansion depends on ionic composition and water availability. Polymers can expand their volume up to 1000 times to form the framework of the mucus gel [35].

On the other hand, intestinal transmembrane mucins (MUC1, MUC3, MUC4 and MUC13) are intercalated in the apical surface of the intestinal epithelium forming the glycocalyx layer [32]. In contrast to the sterile inner layer of mucus, the outer mucus layer is rich in gut bacteria [29]. These bacteria use diet fiber as energy source; however, under a fiber-free diet they consume MUC2 polysaccharides, leading to a thinner inner mucus layer and dysbiosis [36], as well as bacteria penetration into the lamina propria contributing to IBD development [18].

### 2.2. Mucus Layer under Inflammatory Conditions

The stability of the mucus layer is crucial for intestinal homeostasis, in which MUC2 is secreted at a basal rate. This secretion can be influenced by mediators as cytokines, microbial products, autophagic proteins, reactive oxygen species and inflammasome components [37,38]. Commensal and pathogenic bacteria can regulate mucin production [28]. In the small intestine, a continuous basal secretion of mucus creates a flow towards the lumen that, together with antibacterial agents, keeps microorganisms away from the epithelial surface. Antibacterial agents are secreted by Paneth cells and enterocytes of the crypt bottom. On the other hand, in the colon, the inner mucus layer is the first line of defense against bacteria [39].

The mucus layer is a natural and selective habitat for the gut microbiota [40], which in turn influences mucus composition and may promote mucus secretion and increase mucus layer thickness [41]. Therefore, the gut microbiota affects mucus layer function, possibly through specific bacteria that shape the glycan profile of the mucus, although molecular details remain incompletely identified [42].

There is high number of enteropathogens that have evolved mechanisms to penetrate the mucus barrier. Most of them produce a kind of serine proteases that cleave glycoproteins such as mucins [43]. Moreover, cytokines are involved in the inflammatory response and regulate many cellular and molecular processes including mucus production. In this regard, TNF-α and IL-1β, which are implicated in inflammatory diseases, stimulate gel-forming mucins [43]. Th2 cytokines are implicated in mucin gene expression up-regulating MUC2 and MUC5AC by binding to IL-4 receptor. Endoplasmic reticulum stress in goblet cells produce immature mucins that trigger inflammation [44,45], whereas IL-10 has been found to inhibit endoplasmic reticulum stress and promote intestinal mucus production [43,46].

MUC2 knockout mice show colonization of gut epithelium by enteric pathogens [47,48]. These results suggest that the principal mucus function is to protect the gut against microbes. Binding to mucin oligosaccharide chains likely contributes to immobilize bacteria and prevents them from damaging the intestinal epithelium. MUC2 has also immune roles; small intestine goblet cells provide the passage of soluble luminal antigens by transcytosis. These low molecular weight antigens are delivered to underlying CD103^+^ dendritic cells and may favor IgA production and expansion of regulatory T cells, thereby driving gut homeostasis and tolerance [49]. The commensal microbiota, through its relationship with mucus, prevents colonization by pathogens. In this regard, when antibiotics perturb the gut microbiota, niches are opened facilitating disease development. The gut microbiota also breaks down short-chain fatty acids (SCFA) including acetate, propionate and butyrate [50]. Since butyrate regulates MUC2 production, the microbiota is also involved in the homeostasis of the protective mucus layer [51].

Mucin composition is altered in IBD and mucin structural changes play an important role in IBD onset [52,53]. In fact, alterations of mucus barrier and mucins are observed at IBD onset; goblet cell pathology is a hallmark of UC and CD [43]. Recently, it has been observed that the reduced mucus layer in UC is due to a reduction in the number and secretory function of goblet cells because of an inflammatory environment and due to changes in mucin secretion that persist in the absence of inflammatory cells [54].

The mucus layer is thinner in UC than in the healthy colon, while goblet cell depletion and altered MUC2 glycosylation can be also observed; in addition, MUC2 is undersulfated, weakening mucin protective function [55,56,57]. Despite these results, the expression pattern of MUC2 in UC is not clear. Conversely, MUC5AC, is consistently increased during inflammation in UC [58,59] and its reduced expression is associated with endoscopic improvement in these patients [60]. In Muc5ac^−/−^ mice with DSS colitis, there is an increase in bacterial-epithelial contact and neutrophil recruitment to the colon, therefore, the loss of Muc5ac may exacerbate injury and inflammation in experimental murine colitis [61]. This study also showed a significant increase in MUC5AC/Muc5ac expression during colonic inflammation in biopsies from UC patients and DSS-induced mice colitis [61].

In contrast, mucus thickness is normal or greater than normal in CD, maybe due to goblet cell hyperplasia or increased MUC2 expression, although with a 50% reduction in oligosaccharide chain length [62]. Hence, several changes in the mucosal barrier underlie the complex pathology of IBD.

## 3. Gut Microbiota and the Mucus Layer in IBD

The microbiome plays key roles in the development of mucosal immune responses, pathogen resistance and nutrient metabolism. This fact is in part due to the interaction of the microbiota with components of the mucus layer and the IECs underneath following mucus breakdown. The outer penetrable mucus layer is, thus, the natural habitat for many commensals as they use the exposed mucin glycans for both nutritional support and as attachment sites for bacterial adhesins [63]. Bacteria produce enzymes associated with digestion of different glycans from mucus and fiber from the host diet. Although mucus digestion promotes its physiological turnover and the symbiotic dialogue between the host and commensals such as *Akkermansia muciniphila*, an excessive degradation may be associated to detrimental effects due to epithelial exposure to luminal pathogens [23,36].

Intestinal barrier, antimicrobial and immunomodulatory functions are influenced by several members of the gut microbiota, as recently evidenced in studies with cellular models of the epithelial and mucus layers [64,65]. Some commensals, probiotics, notably *Lactobacillus* and *Bifidobacterium* strains and probiotic mixtures have proved mucus-modulating action not only in IBD-like animal models, but also in gnotobiotic animals as well as animal models of diet-induced obesity, malnutrition and aging (Table 1). In this regard, *Lactobacillus rhamnosus* CNCM I-3690 induces reinforcement of the intestinal barrier against chemical-induced colitis with similar effects to those showed by the well-known beneficial human commensal *Faecalibacterium prauznitzii* A2-165 [66]. *F. prausnitzii* is a physiological sensor of gut health and exerts a complementary action with *Bacteroides thetaiotaomicron* as acetate consumer and butyrate producer to balance the mucus barrier by modifying goblet cell differentiation, mucin gene expression and glycosylation [67]. According to the Human Microbiome Project, *Bifidobacterium dentium*, as other *Bifidobacterium* strains, is a recognized member of the healthy infant and adult human gut microbiota [68] and its beneficial effect in rescuing mucus layer function has been proved in gnotobiotic mice [69].

The presence of *A. muciniphila* within the mucus layer is another control mechanism of host mucus turnover, which is essential to gut barrier function. Despite *A. muciniphila* being known as a mucin-degrading bacterium, high-fat-fed mice supplemented with this bacterium show increased counts of goblet cells and secretion of antimicrobial peptides and acylglycerols involved in intestinal and glucose homeostasis [70]. *A. muciniphila* also restores aging-related thinness of the colonic mucus and alterations in inflammatory and immune mediators [71]. Beyond data obtained in murine models, abundance of this bacterium has been inversely associated with obesity and type 2 diabetes in humans, thereby suggesting a physiological role for this mucus colonizer in the regulation of chronic metabolic and inflammatory disorders [70,72]. 

In addition to bacteria themselves, some microbial components/metabolites, such as pathogen-associated molecular patterns and SCFA as well as bacterial metabolites of dietary fiber, can also act on the mucus barrier [39,83]. For example, this is the case of specific outer proteins from *A. muciniphila* [72], or polysaccharide A from *Bacteroides fragilis* [84], which are sensed by Toll-like receptors and ultimately influence host immunity. SCFA, in addition to their roles as energy source for the epithelium and inducers of immune tolerance through T-regulatory cells, are able to stimulate both the discharge of intestinal mucins and MUC2 gene expression [85]. Moreover, it has been suggested that the beneficial effects of *Escherichia coli* Nissle 1917 treatment on chemical-induced colitis (Table 1) may be transferable to germ free mice, but to a lower extent, via fecal microbiota transplantation after mucosal colonization and restoration of the inflammatory responsiveness [76].

The disruption of barrier function in response to IBD or mucosal stressors such as nonsteroidal anti-inflammatory drugs has been addressed in relation to the activity of microbial species on human gut permeability [86]. Diverse probiotics, in particular combination of agents such as the probiotic mixture VSL#3, which has shown beneficial effects on mice colitis (Table 1), have been evaluated in humans by placebo-controlled trials. Pouchitis is one of the intestinal diseases showing increased mucosal permeability. A Cochrane systematic review found that a specific formulation of VSL#3 was superior to placebo in maintaining pouchitis clinical remission at 9–12 months of follow-up, but neither *Lactobacillus* GG nor *Bifidobacterium longum* resulted in clinical improvements at 12 weeks and 6 months, respectively [87]. However, the evidence on this topic obtained by randomized clinical trials still presents some methodological limitations and is not supported by high-quality clinical studies [88,89]; hence, further research is warranted.

## 4. Dietary Compounds and the Mucus Layer in IBD

Dietary factors need to be considered when evaluating the complex relationship between the host, microbiota and the mucus layer. Dietary patterns and specific foods or nutrients may affect the gut barrier directly or indirectly by shaping microbial species known to influence mucosal protection and inflammatory processes [90]. Hence, Western diet and low-grade inflammation are interlinked factors associated with a growing number of immune-mediated inflammatory diseases such as IBD [91].

Diet is mainly composed of macronutrients including proteins, lipids and carbohydrates and micronutrients as vitamins and minerals. Some dietary factors may increase intestinal permeability and consequently contribute to barrier dysfunction in IBD, while others may reinforce the gut barrier [86]. The influence of the different food compounds in the mucus barrier has been evaluated in animal models, both in health and IBD-like models (results summarized in Table 2). Total proteins and specific protein hydrolysates and bioactive peptides from both animal and vegetable sources can affect the gastrointestinal barrier protecting against experimental IBD through modulation of the levels of mucus and IECs constituents, pro/anti-inflammatory markers, antioxidant enzymes, immune mediators and microbiota communities [92]. However, regardless of the protein sources, disruption of the intestinal crypts, number of goblet cells and protein and gene expression of Muc2 has been reported in mice fed with a high-fat diet [93].

High-fat diet has been recently linked to the impairment of mucus layer and stimulation of epithelial oxidative stress and apoptosis, as well as induction of barrier-disrupting molecules and bacterial species [111]. Consistent with this observation, previous studies associated Western diets characterized by animal fat and proteins, sugars and processed food to higher *Bacteroides* and lower *Prevotella* populations, while the Mediterranean diet rich in fruits, vegetables, nuts and whole grains shifted toward abundance of *Prevotella* and fiber-degrading bacteria along with increased production of SCFA [112,113]. Likewise, as pointed by a recent analysis of the relation between dietary factors and the microbiome of healthy volunteers and IBD patients, processed and animal foods are associated with increased abundances of *Firmicutes* and *Ruminococcus* species, but plant foods and fish positively influence SCFA-producing commensals and restrain pathobionts, diet thereby influencing a characteristic microbial environment of intestinal inflammation [114]. Furthermore, high-fat diet also drives colorectal tumorigenesis in mice via intestinal dysbiosis, metabolite dysregulation and gut barrier dysfunction [115].

In addition to the high-fat content, it should be considered that other factors of the Western diet such as a low fiber content may contribute to the negative effects on inflammation. Dietary fiber enriches the gut environment and provides a rich niche for microbial growth to those species able to utilize fiber subtracts [116]. Most bacteria preferentially choose the non-digested food polysaccharides as energy source. Therefore, in fiber-deficient diets, common in the western population, gut bacteria depend to a greater extent on less favorable substrates, especially dietary and endogenous proteins and mucus glycoproteins [42,108]. Mucin glycans are catabolized through a sequential action of different microbial enzymes such as carbohydrate-active enzymes [117]. The degradation of host mucins could negatively impact on mucus homeostasis and enhance pathogen susceptibility [39,63]. This microbial activity may also lead to increased production of harmful metabolites derived from the fermentation of amino acids that contributes to mucus degradation and chronic diseases [36]. Fiber-rich diet is likely suggested to counteract protein fermentation, hence ameliorating the non-desired effects of meat and fats [116].

The preventive effect of fiber may be associated to increased production of SCFA [118], which enhance mucus and antimicrobial peptides secretion, modulate immune function and oxygen levels and reinforce epithelial tight junctions [116,119]. Indeed, some studies in mice have shown that supplementation of high-fat diets with fiber alleviate many of the adverse effects on the mucous barrier (main outcomes summarized in Table 2) in parallel to modulation of microbial composition and SCFA production. The animal models displayed intestinal alterations because of both western style diet-induced obesity and chemical-induced colitis [77,108,109,110]. Particularly, low amount of the prebiotic fiber inulin (1% supplementation in the drinking water) has been shown to correct the penetrability of the inner mucus layer and complement the favorable effects of probiotic *B. longum* on mucus growth [77]. Moreover, soluble inulin (20% fiber supplementation in the high-fat diet), but not insoluble cellulose, prevented microbiota encroachment and further improved gut health by resolution of metabolic alterations, adiposity and glycemic control [108]. On the contrary, the ratio between high-simple sugars/low-fiber contents in diet would predispose the activity and abundance of mucin-degrading microbiota and, in a long-term, the dysfunction of the gut barrier and subsequent inflammation [117]. Noteworthy, a recent systematic review with meta-analysis has found that the intake of dietary fiber is lower in adults with IBD in comparison to healthy individuals [120].

Beyond macronutrients, the essential role of micronutrients [121] and other dietary compounds as fatty acids [122] and phytochemicals [123] at regulating mucosal inflammation and microbiome in IBD has been recently reviewed. On the other hand, some food additives as emulsifiers, maltodextrins and carrageenan may induce increased intestinal permeability, mucus thinness and alterations in the gut microbiota associated with gut barrier dysfunction and negative effects on IBD [39,124].

Evidence from human dietary intervention studies on this topic is still limited. A few human trials have evaluated the effect of some prebiotics and symbiotics on the improvement of intestinal permeability, although most of them found only marginal or non-significant differences compared to placebo [86]. The impact of fiber on human mucus barrier is variable depending on factors such as the study population, the gastrointestinal location and the type of fiber [90]. However, a systematic review by Leech and collaborators did not identify lower fiber consumption as a risk factor for intestinal permeability [125]. On the contrary, within a Western-style diet, fat intake and either inadequate protein intake or excess animal-derived protein are suggested as independent risk factors for altered intestinal integrity [125]. A dietary intervention study found that different animal and non-meat protein sources had modest effects on the abundance of mucosa-associated microbial taxa, these effects being, however, less marked when compared to the impact of the level of saturated fats [126]. Likewise, high-fat diets negatively correlate with microbial diversity, richness and abundance of *F. prausnitzii* and *A. muciniphila* and are associated with reduced bacterial load in human fecal samples [127,128].

## 5. Conclusions

Mucosal barriers represent the first physical host defensive mechanism. They not only keep microorganisms away from the epithelium preventing microbial translocation into mucosal tissues, which would trigger exacerbated inflammatory-immune responses, but also provide a rich source of nutrients for commensals. The gastrointestinal mucus, mostly composed of mucins, plays a vital role in the proper function of the digestive tract and accordingly in human health. Hence, alterations in mucus composition, organization, secretion and degradation or its functionality are linked to a variety of diseases including IBD. A multifactorial model is proposed for IBD pathogenesis where several alterations converge and involve an intestinal barrier failure along with the dysregulation of the immune system. It still remains unclear whether mucus alterations are cause or consequence of the disease. Moreover, scientific interest on host-microbiome interactions displayed at the gastrointestinal mucus layer has increased over the last years, providing evidence that has sharply improved our knowledge on how microbiota regulates host health. Of note, gut lumen environmental factors including gut microbiota and dietary compounds and the complex three-way interaction between both elements and the mucus layer, may act on gut barrier integrity and regulate a healthy gastrointestinal homeostasis, as opposed to IBD alterations (Figure 1).

Indeed, in general terms, the studies summarized in the present review suggest that some microbiota/diet interactions play a role in maintaining gut homeostasis and mucus function. However, current research on these topics presents several limitations and some questions remain open, especially regarding the deficiencies in the design of robust clinical trials and long-term, evidence-based studies to implement findings in practice. Considering the multifactorial nature of IBD and the lack of effective therapies to cure the disease, improving our understanding on the regulation of the intestinal mucus barrier should be further considered with the goal of providing help in IBD management.

## Figures and Tables

**Figure 1 ijms-22-10224-f001:**
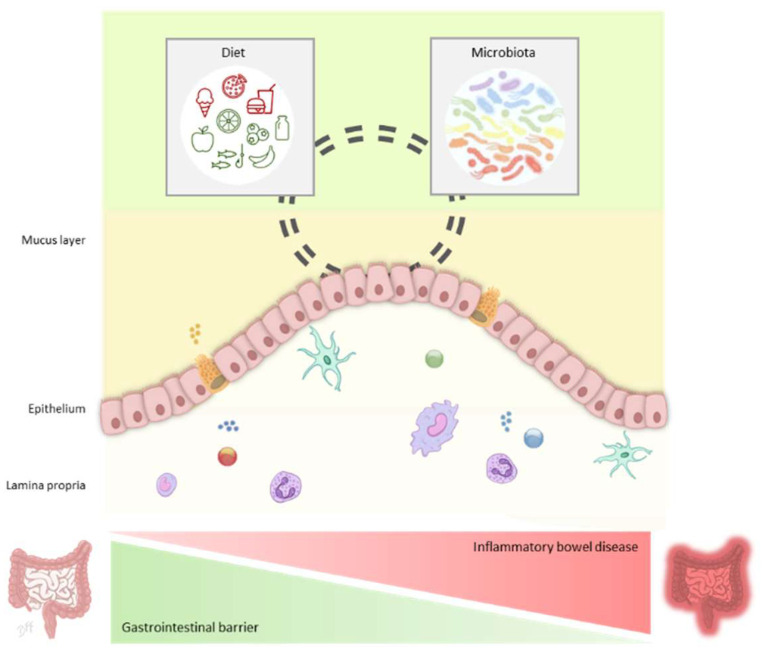
Diet and the gut microbiota regulate gastrointestinal barrier in healthy gut and inflammatory bowel disease. Schematic representation of the influence of diet that may act directly on components of the gastrointestinal barrier and indirectly through shaping microbiota composition, function and its energy source. Some dietary compounds usually found in Mediterranean diet (green) may favor the gastrointestinal barrier, as opposed to the factors of Western-style diet (red). The impact of host-microbiota interactions at the gut lumen and mucus layer, epithelium and mucosal immune system is essential to balance the gastrointestinal barrier in contrast to the alterations underlying inflammatory bowel disease.

**Table 1 ijms-22-10224-t001:** Summary of studies evaluating in animal models the effects of gut bacterial species on the mucus layer.

Bacterial Strain	Animal Model	Experimental Administration	Study Period	Outcomes and Mechanisms of Action	Reference
*Lactobacillus rhamnosus* CNCM I-3690 and *L. paracasei* CNCM I-3689	DNBS-induced colitis in C57BL/6J mice	Intragastric administration with 1 × 10^9^ CFU/mL	10 days	− Restoration of the induced increase of the colonic permeability by *L. rhamnosus* CNCM I-3690 but not *L. paracasei* CNCM I-3689. − Reinforcement of the intestinal barrier by modulation of the expression of epithelial tight junction proteins and reduced colonic levels of IL-4, IL-6 and IFN-γ cytokines.	[66]
*Lactobacillus rhamnosus* CNCM I-3690	DNBS-induced colitis in C57BL/6J mice	Intragastric administration with 5 × 10^9^ CFU/mL	10 days	− Improvement of colonic macroscopic scores, colonic cytokine levels, colon and ileum myeloperoxidase activity and intestinal permeability. − Increase in the contents of acid and neutral mucopolysaccharides in goblet cells and MUC2 staining in the mucus layer. − Induction of an anti-inflammatory response in the spleen and mesenteric lymph nodes. − Upregulation of genes involved in gut health and protective functions against permeability, analyzed by colonic transcriptome analysis.	[73]
*Lactobacillus reuteri* R2LC and *Lactobacillus reuteri* 4659	DSS-induced colitis in C57BL/6J mice	Oral gavage with 1 × 10^8^ live bacteria	14 days	− Reduction of colitis clinical and histological severity indexes. − Reduction of the pro-inflammatory markers myeloperoxidase, IL-1β, IL-6 and mouse keratinocyte chemoattractant. − Induction of adherent mucus thickness and expression of tight junction proteins occludin and ZO-1 in the colonic crypts.	[74]
*Bacillus subtilis* JNFE0126	DSS-induced colitis in C57BL/6J mice	*B. subtilis*-fermented milk oral gavage (6 × 10^8^ CFU/mL)	21 days	− Prevention and alleviation effects against intestinal inflammation in both the active and recovery phases. − Reduction of disease activity index and pathological changes in the small intestine and colon. − Amelioration of neutrophil infiltration and mucosal pro-inflammatory cytokines. − Promotion of the proliferation of intestinal stem cells (*Lgr5*), epithelial cells (*CDx2*) and mucosal barrier (*Mucin2, Zo-1, Villin*). − Increase of microbiota diversity and restoration of gut balance.	[75]
*Escherichia coli* strain Nissle 1917	DSS-induced colitis in BALB/c mice	Intragastric administration with 1 × 10^9^ CFU/mL	17 days	− Protection against induced clinical and histopathological colitis and preservation of intestinal permeability. − Reduction of mucosal infiltration of neutrophils and eosinophils, myeloperoxidase activity and IL-1β and CXCL1/KC levels. − Expansion of regulatory T-cells in the Peyer´s patches	[76]
*Bifidobacterium longum* NCC 2705	Western style diet-induced obesity in C57BL/6J mice	Supplementation of the drinking water with 2 × 10^6^ CFU/mL	4 weeks	− Alteration of gut microbiota composition with loss of *Bifidobacterium* taxa and reduced growth rate and higher penetrability of the colonic mucus by the Wester style diet. − Prevention of mucus growth defects in the probiotic-supplemented group.	[77]
*Bifidobacterium dentium* ATCC 27678	Swiss Webster germfree mice	Oral gavage with 2 × 10^8^ CFU/mL	1–2 weeks	− Microbial colonization of the colon mucus layer in gnotobiotic mice. − Increase in the number of filled intestinal goblet cells and modulation of mucus glycosylation. − Promotion of cell maturation and function with increased expression of *Muc2,* Krüppel-like family of zinc-finger transcription factor 4 (*Klf4*), resistin-like molecule-β (*Relm*-β) and trefoil factor 3 (*Tff3*), without corresponding changes in mucin-modulating cytokines.	[69]
*Lactobacillus reuteri* LR6	Protein and energy malnutrition in *Swiss* mice	Diet with fermented product or bacterial suspension at 1 × 10^9^ CFU/day	1 week	− Reinforcement of intestinal health. − Expansion of the intact morphology of colonic crypts and lamina propria, normal goblet cells, while lessening of inflammation in large intestine and spleen and absence of fibrosis. − Stimulation of secretory IgA^+^ cells and the counts of phagocytic macrophages and bone marrow derived dendritic cells.	[78]
*Akkermansia muciniphila* Muc^T^ BAA-835	Accelerated aging *Ercc1^-/Δ7^* mice	Oral gavage with 2 × 10^8^ CFU/200 µL	10 weeks	− Expansion of colonic mucus thickness. − Decrease in the expression of colonic and ileal genes related to inflammation and immune and metabolic functions. − Lower presence of B cells in colon, decreased frequencies of activated CD80^+^CD273^−^ B cells in Peyer’s patches and Ly6C^int^ monocytes in spleen and mesenteric lymph nodes. − Expansion of mature and immature B cells in bone marrow and peritoneal resident macrophages.	[71]
VSL#3 probiotic mixture	DSS-induced colitis in *Muc2^−/−^* mice	Oral gavage with 2.25 × 10^9^ CFU/day	2 weeks	− Improvement of compromised intestinal barrier without significant protection against colitis progression. − Attenuation of basal pro-inflammatory cytokine levels and induced production of innate cytokines and reactive oxygen species. − Enhancement of tissue regeneration growth factors, antimicrobial peptides and abundance of bacterial gut commensals. − Enhanced production of SCFAs, mainly acetate.	[79]
VSL#3 probiotic mixture	DSS-induced colitis in C57BL/6J mice	Oral gavage with 3 × 10^9^ live bacteria	60 days	− Anti-inflammatory effect with reduced scores of disease activity index, histological activity index and myeloperoxidase activity. − Reduction in IgM, IgG and IgA levels in colon mucus and the number of T follicular helper cells in mesenteric lymph nodes.	[80]
*Lactobacillus johnsonii* IDCC9203, *Lactobacillus plantarum* IDCC3501 and *Bifidobacterium animalis* subspecies *lactis* IDCC4301 (ID-JPL934 probiotic mixture)	DSS-induced colitis in BALB/c mice	Oral gavage with probiotic mixture (1 × 10^6^–1 × 10^9^ CFU/day)	8 days	− Dose-dependent reduction of colitis symptoms including body weight loss, diarrhea and bloody feces and colon length contraction. − Similar effects to sulfasalazine at 500 mg per kg per day. − Suppression of the infiltration of immune cells into mucosa and submucosa, crypt damage, expression of pro-inflammatory TNFα, IL-1β and IL-6. − Restoration of physiological epithelial cells and goblet cells histology.	[81]
*Lactobacillus rhamnosus, L. acidophilus and Bifidobacterium bifidumi*	High fat diet-induced obesity in *Swiss* mice	Oral gavage with probiotic mixture (6 × 10^8^ CFU of each strain; final concentration of 1.8 × 10^9^ CFU of bacteria)	5 weeks	− Induction of gut microbiota alterations, intestinal permeability, LPS translocation and systemic low-grade inflammation, reverted by the probiotic mixture. − Endorsement of glucose tolerance, hyperphagic behavior, hypothalamic insulin and leptin resistance.	[82]

DNBS: dinitrobenzene sulfonic acid; DSS: dextran sulfate sodium; CFU: colony-forming units; SCFAs: short-chain fatty acids; LPS: lipopolysaccharide.

**Table 2 ijms-22-10224-t002:** Summary of studies evaluating in animal models the effects of food compounds on the mucus layer.

Food Group/Compounds	Animal Model	Experimental Administration	Study Period	Outcomes and Mechanisms of Action	Reference
*Proteins*					
Total proteins	Adult finishing pigs	Three study groups (16%, normal dietary protein concentration; 13%, low dietary protein concentration; 10%, extremely low dietary protein concentration)	50 days	− Decrease in ileal bacterial richness, levels of intestinal SCFAs and biogenic amines with reduction of protein concentration. − Inhibition of stem cell proliferation, decrease in the expression of biomarkers of intestinal cells (Lgr5 and Bmi1) and alteration of gut bacteria community and ileal morphology in the 10% protein group. − Improvement of ileal and colonic bacterial community and enhancement of tight junction proteins (occludin and claudin) and ileal barrier function in the 13% protein group.	[94]
Total proteins	Growing pigs	Three study groups (18%, normal dietary protein concentration; 15%, low dietary protein concentration; 12%, extremely low dietary protein concentration)	30 days	− Decrease in the levels of most bacterial metabolites with reduction of protein concentration. − Reduction of ileal barrier function and tight junction proteins (occludin, zo-3, claudin-3 and claudin-7) in the 12% protein group. − Deficit in the development of intestinal villi and crypts and increased intestinal LPS-permeability in the low protein groups. − Enhancement of ileal richness, bacterial diversity and expression of intestinal stem cells (Lgr5) in the 15% protein group.	[95]
Chicken and soy proteins	C57BL/6 mice	Chicken or soy protein-based diets	4 weeks	− Increase in the thickness of the colonic mucus layer, the number of goblet cells, the expression of Muc2 mRNA and the abundance of *A. muciniphila*, in comparison to soy-protein-based diet.	[96]
Milk casein	Rats	Milk casein hydrolysate	8 days	− Stimulation of terminal ileal endogenous nitrogen flow. − Upregulation in the expression of mucin genes Muc3 in the small intestine and Muc4 in the colon.	[97]
Milk casein	Zucker rats	Milk casein hydrolysate	8 weeks	− Rise in the secretion of O-linked glycoproteins in the fecal material. − Upregulation in the expression of mucins genes (Muc3 and Muc4) in the ileal and colonic intestinal regions.	[98]
Milk β-casein	Rats pups	Milk β-casein peptide f(94–123)	9 days	− Increase in the number of goblet cells and crypts containing Paneth cells in the small intestine. − Upregulation in the expression of intestinal mucins (Muc2 and Muc4) and antibacterial factors (defensin-5 and lysozyme).	[99]
Milk β-casein	Indomethacin-induced jejunal injury in rats	Milk β-casein peptide f(94–123)	8 days	− Preventive amelioration of macroscopic and microscopic intestinal damage. − Preventive reduction of goblet cells, increased myeloperoxidase activity and expression of TNF-α and active caspase-3.	[100]
Goat whey	DNBS-induced colitis in CD1 mice	Goat whey proteins, fatty acids and oligosaccharides	16 days	− Reduction of colitis activity index and symptoms and mucosal leukocyte infiltration. − Downregulation in the expression of pro-inflammatory IL-1β, IL-6, IL-17, TNF-α, iNOS, MMP-9 and ICAM-1. − Increase in barrier function and upregulation in the expression of Muc2, Muc3, occludin and zonula occludens-1.	[101]
Hen egg	DSS-induced colitis in piglets	Egg white lysozyme	5 days	− Restoration of colitis symptoms, mucosal inflammation, muscle wall thickening, gastric permeability and mucin gene expression. − Down-regulation of intestinal expression of pro-inflammatory TNF-α, IL-6, IFN-γ, IL-8 and IL-17 and up-regulation of tolerogenic TGF-β and Foxp3.	[102]
Soybean protein	DSS-induced colitis in piglets	Soybean protein derived di- and tri-peptides	5 days	− Decrease in gut permeability, crypt elongation and muscle thickness, colonic expression of pro-inflammatory mediators and myeloperoxidase activity. − Down-regulation of ileal mRNA levels of IFN-γ, TNF-α, IL-12B and IL-17A and up-regulation of FOXP3 expression.	[103]
Pea protein	DSS-induced colitis in C57BL/6J mice	Pea seed protein extracts	23 days	− Amelioration of colitis-induced histological alterations. − Restoration of colonic protein levels related to epithelial barrier function and mRNA expression of pro-inflammatory cytokines, inducible enzymes, metalloproteinases, adhesion molecules and toll-like receptors. − Gut modulation of bacterial abundances towards healthy conditions.	[104]
*Lipids*					
High- and low-fat diets	C57BL/6J mice	Chicken, soy or pork protein-based administration either with low fat (12% kcal) or high fat (60% kcal) diets	12 weeks	− Disruption of crypt depth, numbers of goblet cells and protein and gene expression of Muc2 in the high-fat diet group, regardless of protein diets. − Upregulation of Muc2 gene expression by meat proteins in the low-fat diet group. − Reduction of intestinal barrier, zonula occludens-1 and E-cadherin proteins and increase of colonic IL-1β expression and serum TNF-α and IL-6 by meat proteins in the high-fat diet group.	[93]
High-fat diet	C57BL/6 mice	High-fat diet (56.7 Fat kcal %), in comparison with normal chow diet (12.0 Fat kcal %)	8 weeks	− Reduction of fecal weight, increase of total gastrointestinal transit time and colon transit time and reduction of colonic mucus in the high-fat diet group	[105]
High-fat diet	Spontaneous colitis in *Winnie* mice	High-fat diet (46% available energy as fat), in comparison with normal chow diet (11% available energy as fat)	9 weeks	− Increase in diarrhea scores, bloody feces, more severe and widespread colonic damage with prominent mucosal erosions and crypt abscesses. − Induction of endoplasmic reticulum stress (*Grp78* and *sXbp1*) and oxidative stress (*Nos2*) markers. − Down-regulation of goblet cell differentiation (*Klf4*) and intestinal claudin-1 protein staining.	[106]
Flaxseed oil	LPS-induced intestinal injury in weaned piglets	Supplementation of diets with flaxseed oil in comparison with corn oil (5% weight:weight)	3 weeks	− Restoration of intestinal morphology, jejunal lactase activity, necroptosis signals and claudin-1 protein expression. − Down-regulation of mRNA expression of intestinal toll-like receptors 4 (TLR4), myeloid differentiation factor 88 (MyD88), nuclear factor kappa B (NF-κB), nucleotide-binding oligomerization domain proteins (NOD1, NOD2) and receptor-interacting protein kinase 2 (RIPK2). − Increased levels of intestinal α-linolenic acid, eicosapentaenoic acid and total n-3 polyunsaturated fatty acids.	[107]
*Fiber*					
Inulin	Western style diet-induced obesity in C57BL/6J mice	1% oligofructose-enriched inulin supplementation in the drinking water	4 weeks	− Alteration of gut microbiota composition with loss of *Bifidobacterium* taxa and reduced growth rate and higher penetrability of the colonic mucus by the Wester style diet. − Prevention of the penetrability of the inner mucus layer in the fiber inulin group.	[77]
Inulin and cellulose	Western style diet-induced obesity in C57BL/6J mice	Supplementation of high-fat diets (60 kcal% fat) with 20 % fiber	4 weeks	− Protection against diet-induced low-grade inflammation and metabolic syndrome by fermentable inulin fiber, but not insoluble cellulose fiber. − Restoration of epithelial cell proliferation and colon atrophy, microbiota loads, IL-22 production and antimicrobial gene expression. − Suppression of adiposity and improvement of glycemic control.	[108]
Pectin	TNBS- and DSS-induced colitis in C57BL/6J mice	Diet supplemented with characteristically high (5% orange pectin) in comparison to low (5% citrus pectin) side chain content of pectin	10–14 days	− Amelioration of clinical symptoms and colonic damage. − Decrease in levels of colonic IL-1β and IL-6. − Increase in the fecal concentration of propionic acid. − Protective effects against intestinal inflammation even in mice treated with antibiotics.	[109]
Microbiota-accessible carbohydrates	High-fat and fiber-deficient diet in C57BL/6J mice	Supplementation of high-fat (31.5% fat by weight) and fiber-deficient (5% fiber by weight) diet with microbiota-accessible carbohydrates	15 weeks	− Improved intestinal barrier function by increased colonic mucus thickness and tight junction protein expression. − Amelioration of endotoxemia, colonic and systemic inflammation and enhancement of microbiota richness and α-diversity. − Improvement of cognitive impairment via the gut microbiota-brain axis.	[110]

DNBS: dinitrobenzene sulfonic acid; DSS: dextran sulfate sodium; LPS; lipopolysaccharide; TNBS: 2,4,6-trinitrobenzenesulfonic acid.

## Data Availability

All data described in the review are included in this published article.

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
