# Peer review of "Gut Microbiota and Dietary Factors as Modulators of the Mucus Layer in Inflammatory Bowel Disease"

_ijms, 2021, doi:10.3390/ijms221910224_

Round 1
Reviewer 1 Report
In the present review titled as "Gut microbiota and dietary factors as modulators of the mucus 2 layer in inflammatory bowel disease", the authors aimed to discuss the current limitations on this topic, especially regarding the design of robust human trials, and to highlight the potential interest of improving our understanding of the regulation of the intestinal mucus barrier in IBD.
Overall, the manuscript is well written and provides interesting data on this topic. Moreover, since additional modifications performed following other reviewer's comments, its content has been improved.
Reviewer 2 Report
The authors have significantly edited the manuscript resulting in a more specific and topical review. The authors have successfully addressed all my previous concerns. However, I noted the following in the newly edited section.
- Line 100-102. You quote “IBD integrome” from reference 28. I looked up reference 28, they don’t use that term at all in their paper. Where is the quote from? If it’s of your own making – you need to identify it as such. The construction of the sentence suggests that the authors of ref 28 used that term.